# A model-based analysis of the health impacts of COVID-19 disruptions to primary cervical screening by time since last screen for current and future disruptions

Emily A Burger[1,2]*, Inge MCM de Kok[3†], James F O'Mahony[4†], Matejka Rebolj[5], Erik EL Jansen[3], Daniel D de Bondt[3], James Killen[6], Sharon J Hanley[7], Alejandra Castanon[5], Mary Caroline Regan[1], Jane J Kim[1], Karen Canfell[8], Megan A Smith[8]

[1]Center for Health Decision Science, Harvard T.H. Chan School of Public Health, Boston, United States; [2]Department of Health Management and Health Economics, University of Oslo, Oslo, Norway; [3]Department of Public Health, Erasmus MC, University Medical Center Rotterdam, Rotterdam, Netherlands; [4]Centre for Health Policy & Management, School of Medicine, Trinity College Dublin, Dublin, Ireland; [5]Faculty of Life Sciences & Medicine, School of Cancer & Pharmaceutical Sciences, King's College London, London, United Kingdom; [6]Cancer Research Division, Cancer Council NSW, Sydney, Australia; [7]Hokkaido University Center for Environmental and Health Sciences, Sapporo, Japan; [8]Daffodil Centre, University of Sydney, a joint venture with Cancer Council NSW, Sydney, Australia

*For correspondence:
eburger@hsph.harvard.edu

†These authors contributed equally to this work

‡Karen Canfell receives salary support from the National Health and Medical Research Council, Australia (APP1194679).Karen Canfell is the co-PI of an investigator-initiated trial of cervical cancer screening, Compass, run by the VCS Foundation, which is a government-funded not-for-profit charity. Neither KC nor her institution have received funding from industry for this or any other research project.

**Abstract** We evaluated how temporary disruptions to primary cervical cancer (CC) screening services may differentially impact women due to heterogeneity in their screening history and test modality. We used three CC models to project the short- and long-term health impacts assuming an underlying primary screening frequency (i.e., 1, 3, 5, or 10 yearly) under three alternative COVID-19-related screening disruption scenarios (i.e., 1-, 2-, or 5-year delay) versus no delay in the context of both cytology-based and human papillomavirus (HPV)-based screening. Models projected a relative increase in symptomatically detected cancer cases during a 1-year delay period that was 38% higher (Policy1-Cervix), 80% higher (Harvard), and 170% higher (MISCAN-Cervix) for underscreened women whose last cytology screen was 5 years prior to the disruption period compared with guidelines-compliant women (i.e., last screen 3 years prior to disruption). Over a woman's lifetime, temporary COVID-19-related delays had less impact on lifetime risk of developing CC than screening frequency and test modality; however, CC risks increased disproportionately the longer time had elapsed since a woman's last screen at the time of the disruption. Excess risks for a given delay period were generally lower for HPV-based screeners than for cytology-based screeners. Our independent models predicted that the main drivers of CC risk were screening frequency and screening modality, and the overall impact of disruptions from the pandemic on CC outcomes may be small. However, screening disruptions disproportionately affect underscreened women, underpinning the importance of reaching such women as a critical area of focus, regardless of temporary disruptions.

## Editor's evaluation

This article describes the use of three well-established mathematical models of cervical cancer to estimate the impact of COVID-19-related delays in screening access on cervical cancer incidence and delays in diagnosis. Consistent with previous work and the known biology of cervical cancers, the findings that short delays have relatively small effects on population-level cervical cancer risk are reassuring overall, but the impact of screening interval and screening test performance suggests that existing disparities related to screening access may be exacerbated. These results should be useful for policymakers in planning responses to future pandemics or other sources of sudden restriction of screening availability.

## Introduction

The coronavirus disease 2019 (COVID-19) pandemic continues to impact a wide range of health outcomes. In the initial months of the pandemic in 2020, there were severe disruptions to preventive services, including cervical cancer screening. For example, during acute phases of the pandemic in the United States, 59% of federally qualified health centers stopped cancer screenings completely (*Fisher-Borne et al., 2021*), and electronic health records from 39 organizations spanning 23 states found a 67% decline in mean weekly cervical cancer screening volumes (*Mast and Rio, 2020*). While cancer screening volumes gradually improved (*Mast and Rio, 2020*), mid-June 2020 volumes remained around 30% lower than their pre-COVID-19 levels, and cervical volumes have remained 10% lower 2 years into the pandemic (*Christopher and Joe, 2022*).

The risk of developing cervical cancer depends in part on time since last screen (*Dillner et al., 2008*; *Landy et al., 2020*). Despite US recommendations for primary cervical cancer screening of either 3-yearly cytology or 5-yearly human papillomavirus (HPV) testing (*Curry et al., 2018*), there is heterogeneity in adherence to guideline recommendations where both underscreening and over-screening are observed when comparing behavior to recommendations. For example, in the only population-based registry in the United States prior to widespread primary HPV-based screening, 20% of women were not screened within 5 years (*Cuzick et al., 2014*), which was correlated with race and ethnicity, income level, lower levels of education, and lack of insurance (*Johnson et al., 2020*). Conversely, screening more frequently than recommended has been observed in 66% of insured women (*Wright et al., 2021*).

The impact of service disruptions due to COVID-19 may not have affected all women equally. For women without health insurance or unable to access care, or those who avoid care due to fear of COVID-19, the disruptions may continue. In other countries such as the United Kingdom, 30% of survey respondents elicited fall 2020 reported that they were less likely to attend cervical screening now than before the pandemic (*Wilson et al., 2021*). Although the observed decrease in screening attendance ultimately was smaller than surveyed intentions to screen (*Castanon et al., 2022*), the UK study also found that previous nonparticipation was the strongest predictor of low intentions for future post-pandemic participation.

Rebounds toward pre-pandemic attendance levels in aggregate-level metrics may suggest a successful recovery but could actually mask unexpected disparities in coverage. For example, the same disruption period may differentially impact women due to heterogeneity in their screening history so that the impact is greater for those underscreened compared to those that are screened according to recommended guidelines. It will thus be important to understand the influence of variation in women's past behavior as a contributor to underlying risk when assessing past and ongoing disruptions to screening.

Empirically, decreases in cervical cancer diagnoses in 2020 have been confirmed in the United States (*Kaufman et al., 2020*) and elsewhere (*Ijzerman and Emery, 2020*). Previous model-based analyses have projected that temporary disruptions to cervical cancer screening may result in temporal shifts in cancer detection (initial decreases followed by an increase), yielding small net increases in cervical cancer burden (*Burger et al., 2021*; *Smith et al., 2021*). Such decreases are to be expected in the short run due to the reduction in screening and related investigations and any net increases will only be observed in time. Model-based analyses (*Burger et al., 2021*; *Smith et al., 2021*) have shown that maintaining services for the highest risk women may mitigate the potential secondary impacts of COVID-19 on cervical cancer; for example, prioritizing those in need of diagnostic follow-up (surveillance, colposcopies) or excisional treatment, as well as women whose last primary screen did not

involve a highly sensitive test, such as that for the detection of HPV. Furthermore, short delays to cervical screening services among women with a previous negative HPV result had minor effects on cancer outcomes; however, previous analyses have not explicitly stratified outcomes for women by their prior screening history, that is, time since last screen.

Disease simulation models can help assess the impact of service disruptions and policy responses in advance of empirical data. Models can also quantify health consequences of alternative screening disruption scenarios and isolate complex interactions between temporary screening suspensions for women with different underlying screening histories. These simulations can help inform which women are most vulnerable to COVID-19 disruptions and should be prioritized for targeted recovery activities. Therefore, as part of the Covid and Cancer Global Modelling Consortium (CCGMC), we used three US-contextualized cervical cancer natural history models from the Cancer Intervention and Surveillance Modeling Network (CISNET) consortium (https://cisnet.cancer.gov/) to isolate the health impact of temporary disruptions to *primary screening services only* by time since a woman's last screen and primary screening test modality. Multimodel comparative analyses can demonstrate the validity of findings and test the robustness despite structural differences between the models used. The purpose of this analysis is to provide decision-makers with evidence regarding the potential impact of temporary disruptions to the provision of screening services on cervical cancer incidence, either due to the

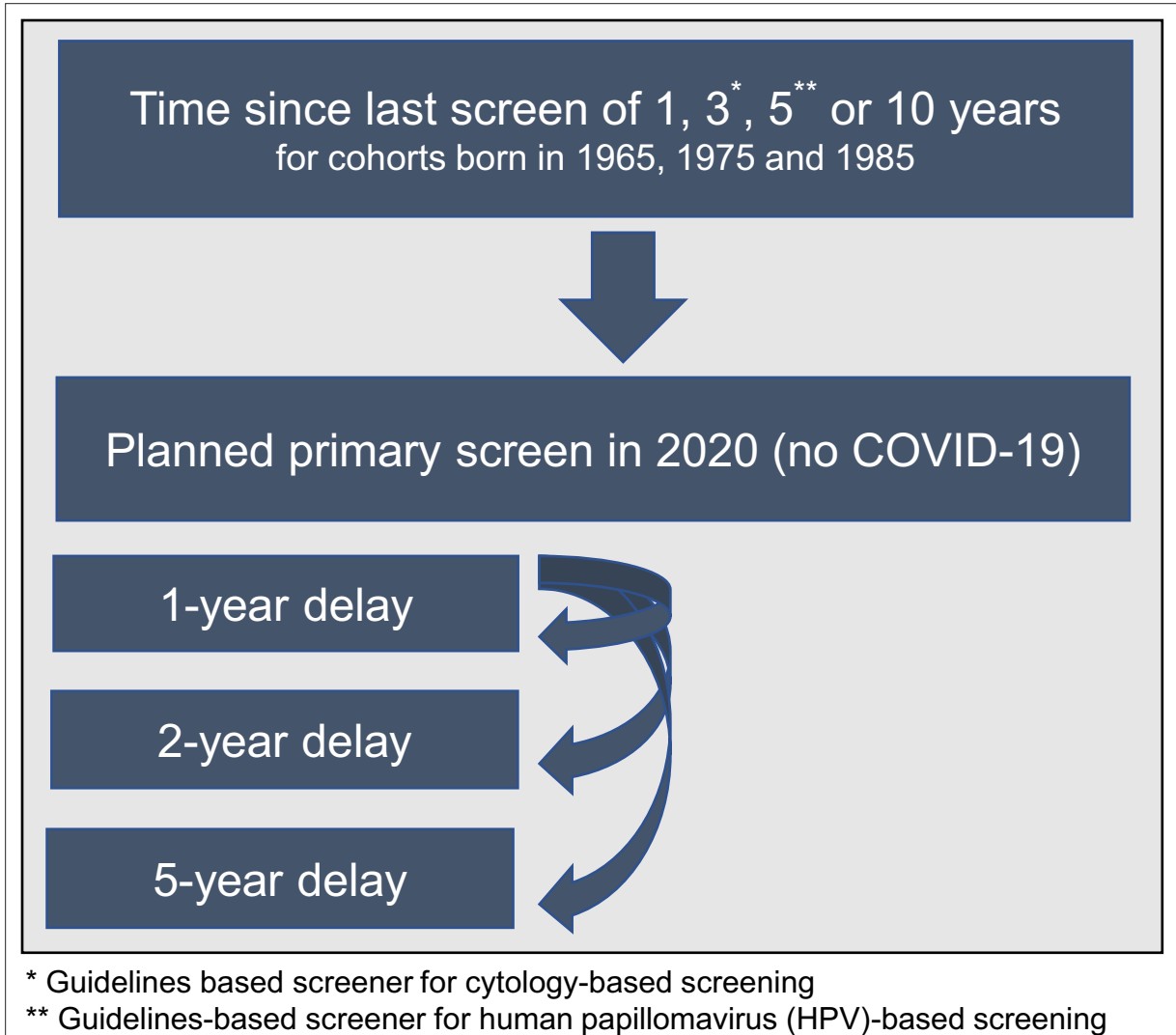

* Guidelines based screener for cytology-based screening
** Guidelines-based screener for human papillomavirus (HPV)-based screening

**Figure 1.** Scenario overview reflecting the heterogeneity in screening history (aligned so that 2020 was 1, 3, 5, or 10 years since their last screen) facing alternative COVID-19 delay disruptions for three birth cohorts of women.

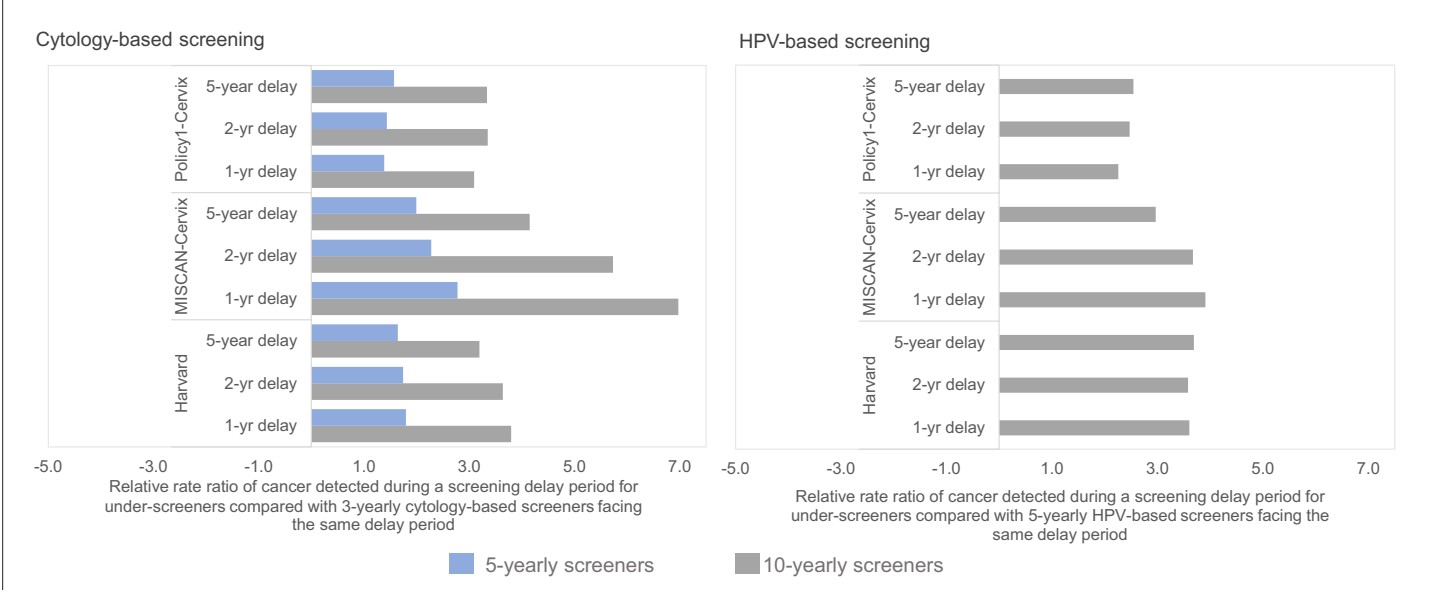

**Figure 2.** Short-term impacts: relative rate ratio of cancer detected during the screening delay period for underscreeners compared with the same delay duration for guidelines-compliant screeners.

COVID-19 pandemic or any other similar disruption, on a disaggregated basis according to women's prior screening history in order to inform any targeted allocation of scarce screening capacity.

## Results
### Short-term impacts

On average, among women aged 35–55 years, the models projected a relative increase in symptomatically detected cancer burden during a 1-year delay period that were higher – 38% higher (Policy1-Cervix), 80% higher (Harvard), and 170% higher (MISCAN-Cervix) – for those who had not screened in 5 years at the time of the disruption, compared with women who attended cytology-based screening according to guidelines (i.e., every 3 years) (*Figure 1*; *Figure 2*, left panels). Compared with guidelines-compliant cytology screeners, the relative excess burden of cancers detected during a 1-year delay period was 3.1 (Policy1-Cervix), 3.2 (Harvard), or 7.0 (MISCAN-Cervix) times higher for women whose last cytology screen was 10 years ago at the time of the disruption. Compared with women who switched to HPV-based screening at age 30 and were guidelines-compliant screening every 5 years, women who screened every 10 years with HPV after age 30 years faced an excess cancer burden that was generally consistent regardless of the disruption period, ranging from 2.2 to 2.5 (Policy1-Cervix), 3.0–3.9 (MISCAN-Cervix), and 3.6–3.7 (Harvard) times higher (*Figure 2*, right panels). Although the relative excess burden among women overdue for screening remained generally similar by delay period, the absolute accumulated rates increased with the length of the delay period (*Appendix 1—table 3*).

### Long-term impacts

The models consistently projected that a 1- or even 5-year temporary disruption to primary screening had a smaller effect on the lifetime risk of developing cervical cancer than the effects from screening frequency and modality we considered (*Figure 3*). For example, within-model comparisons found that the lifetime risk of developing cervical cancer was lower, even in the context of an extreme 5-year screening disruption, for women screened every 3 years with cytology prior to the disruption (Policy1-Cervix [0.21%], Harvard [0.32%], and MISCAN-Cervix [0.32%]) than for women screening every 5 years without a disruption (Policy1-Cervix [0.22%], Harvard [0.42%], and MISCAN-Cervix [0.39%]) (*Figure 3*, upper panels). Set within the wider context of prevention, the models projected that, under an extreme scenario of a 5-year delay, 3-yearly cytology screening maintained nearly all

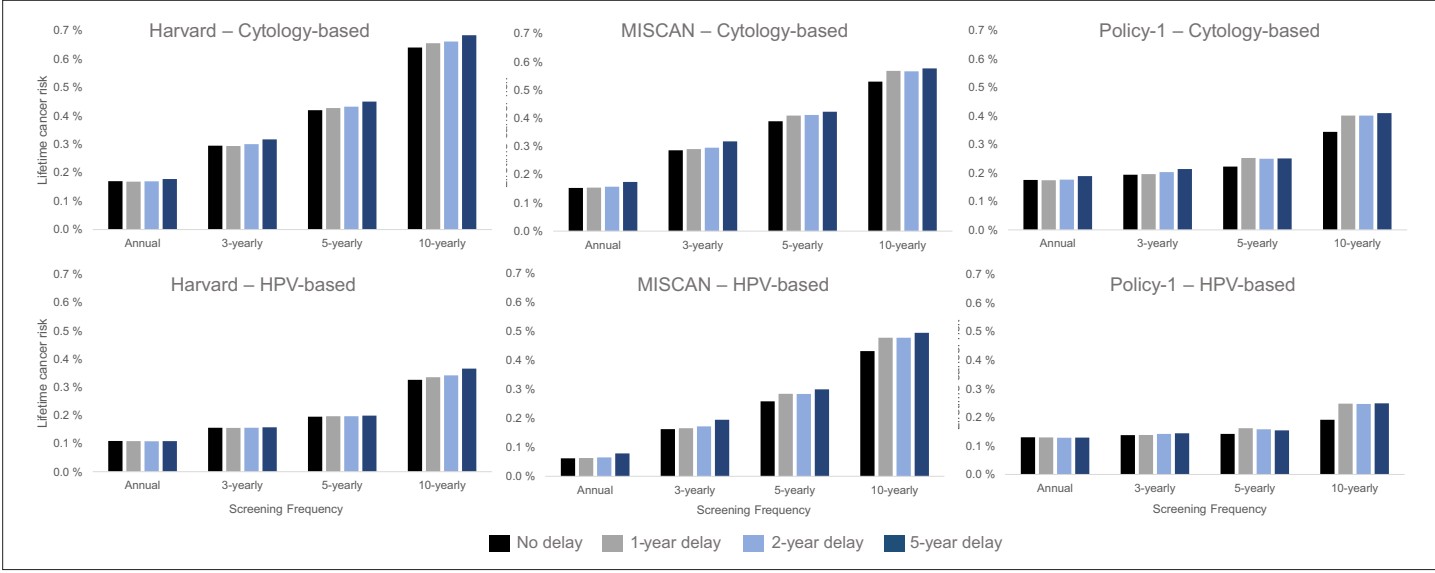

**Figure 3.** Long-term impacts: projected impact of COVID-19-related disruptions to primary cervical cancer screening on the lifetime risk of developing cervical cancer (averaged across the 1965/1975/1985 birth cohorts of women) by time since last screen for cytology-based screening (top panels) and human papillomavirus (HPV)-based screening (bottom panels) for three Cancer Intervention and Surveillance Modeling Network (CISNET-Cervical disease simulation models).

benefits of screening, decreasing from preventing 72.1% to 69.1% (MISCAN-Cervix), 79.9% to 78.4% (Harvard), and 86.5% to 85.1% (Policy1-Cervix) of cancer cases over a woman's lifetime compared with no screening (assuming screening resumed following the disruption) (*Appendix 1—table 4*). In contrast to women screened with cytology over their lifetime, women screened with primary HPV after age 30 years face a lower overall lifetime risk of cervical cancer (and percentage of cancers prevented by screening was higher) compared with cytology-based screening; furthermore, these women generally faced smaller impacts of a COVID-19 disruption to screening relative to screening frequency (*Figure 3*, lower panel; *Appendix 1—table 4*).

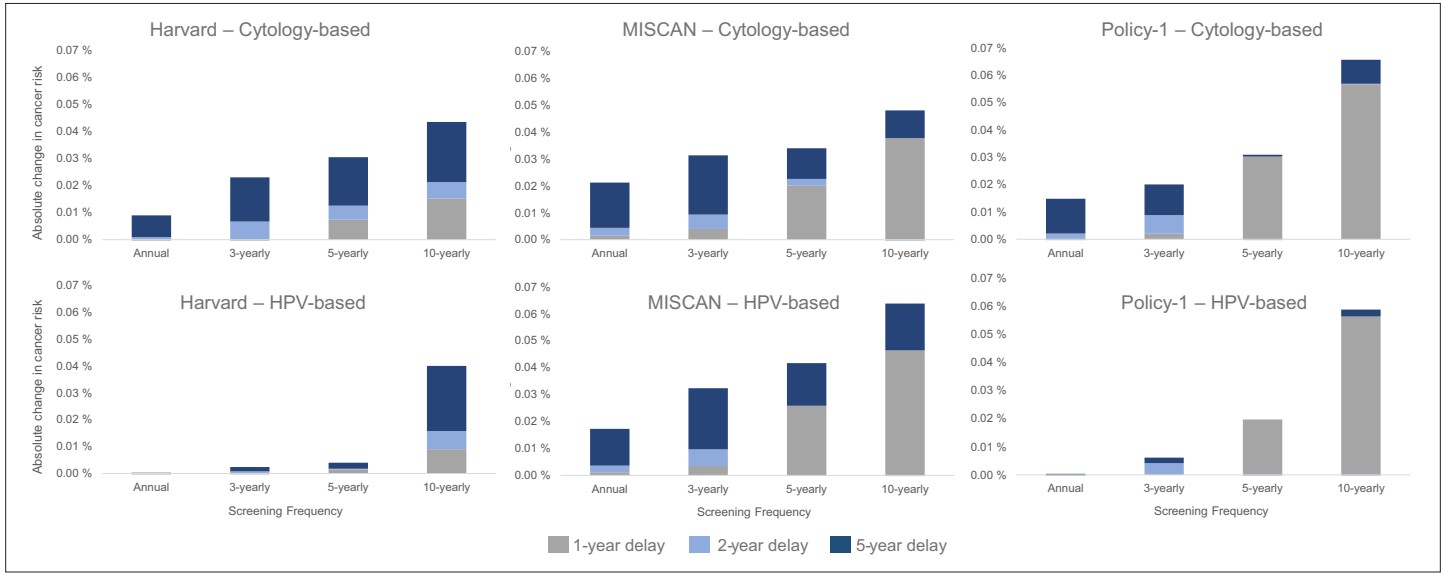

**Figure 4.** Long-term impacts: projected impact of COVID-19-related disruptions to primary cervical cancer screening on the incremental lifetime risk of developing cervical cancer (averaged across the 1965/1975/1985 birth cohorts of women) by time since last screen for cytology-based screening (top panels) and human papillomavirus (HPV)-based screening (bottom panels) for three Cancer Intervention and Surveillance Modeling Network (CISNET)-Cervical disease simulation models.

**Table 1.** Long-term health impacts* of a 5-year temporary delay to screening compared with no delay, by screening history, that is, screening frequency.

| | Screening frequency | | | |
| --- | --- | --- | --- | --- |
| | Annual | 3-yearly | 5-yearly | 10-yearly[†] |
| **Primary cytology-based screening** | | | | |
| *Harvard* | | | | |
| Absolute change in lifetime risk | 0.008% | 0.022% | 0.031% | 0.044% |
| Excess cases over lifetime per 100,000 women | 8 | 22 | 31 | 44 |
| *MISCAN-Cervix* | | | | |
| Absolute change in lifetime risk | 0.021% | 0.031% | 0.034% | 0.047% |
| Excess cases over lifetime per 100,000 women | 21 | 31 | 34 | 47 |
| *Policy1-Cervix* | | | | |
| Absolute change in lifetime risk | 0.014% | 0.020% | 0.029% | 0.066% |
| Excess cases over lifetime per 100,000 women | 14 | 20 | 28 | 66 |
| **Primary HPV-based screening** | | | | |
| *Harvard* | | | | |
| Absolute change in lifetime risk | 0.000% | 0.001% | 0.004% | 0.040% |
| Excess cases over lifetime per 100,000 women | 0 | 1 | 4 | 40 |
| *MISCAN-Cervix* | | | | |
| Absolute change in lifetime risk | 0.017% | 0.032% | 0.041% | 0.063% |
| Excess cases over lifetime per 100,000 women | 17 | 32 | 41 | 63 |
| *Policy1-Cervix* | | | | |
| Absolute change in lifetime risk | 0.000% | 0.006% | 0.012% | 0.058% |
| Excess cases over lifetime per 100,000 women | 0 | 6 | 12 | 58 |

*Risks are rounded to nearest 0.001%.

[†]The women born in 1985 (aged 35 in 2020) received their last screen at age 25 and have not yet made the switch to primary human papillomavirus (HPV)-based screening. In the primary HPV-based analysis, these women would switch to primary HPV-based screening for their remaining lifetime either at 35 (under the no delay scenario) or aged >35 years (with a delay).

Despite the relatively lower contribution of COVID-19-related delays to lifetime risk of developing cervical cancer than screening frequency and test modality, there were important differences in the impact of a delay period on a woman's lifetime risk by the length of time since her last screen and the screening modality used during the last screen (*Figure 4*). In general, annual or 3-yearly screeners faced only nominal excess risks when experiencing a 1-year temporary delay to primary cytology screening, and cancer risks increased disproportionately the longer time had elapsed since a woman's last screen (*Figure 4*, upper panels). For example, all models projected that compared with no COVID-19 delay, an extreme 5-year temporary delay scenario was expected to increase the number of remaining lifetime cervical cancer cases by 20 (Policy1-Cervix), 22 (Harvard), and 31 (MISCAN-Cervix) per 100,000 women screened 3-yearly with cytology, compared to an increase of 44 (Harvard), 47 (MISCAN-Cervix), and 66 (Policy1-Cervix) per 100,000 women screened 10-yearly with cytology (*Table 1*). Importantly, these excess risks for a given delay period were generally lower for HPV-based screeners than for cytology-based screeners (*Figure 4*, lower panels). For example, compared with cytology-based screening, two of the models (Harvard and Policy1-Cervix) found that women screened with primary HPV testing faced smaller excess risks for the same delay duration unless women were screening very infrequently (10-yearly), in which case, the excess risks of cancer

were similar, that is, 44–66 per 100,000 women for cytology versus 40–58 per 100,000 women for HPV 100,000 women.

## Discussion

The results of our comparative health impact modeling study suggest that women who are overdue for routine screening and encounter a further delay face an increased risk of presenting with a symptomatic cancer during the delay period compared with those who attend screening regularly according to guidelines. Furthermore, women who undergo routine guidelines-compliant screening are able to endure a temporary, that is, 1 year, disruption to cervical screening under both primary cytology and HPV-based screening modalities, and those who undergo guidelines-compliant screening with HPV testing are more resilient to longer delays (2 or 5 years). In the United States, where there is heterogeneity in screening behavior (*Wright et al., 2021*), our findings suggest that to minimize population cancer risk, targeted outreach to overscreened and regularly screened women should be a lower priority than outreach to women whose screening history is not up to date. Our findings support outreach to women most vulnerable to COVID-19 disruptions who also faced pre-pandemic barriers to routine screening. Importantly, aggregated metrics demonstrating a near-return to pre-pandemic screening volumes may not be adequate to capture heterogeneities in screening history, and therefore, risk associated with disruptions to screening.

Similar to our previous results (*Burger et al., 2021*), we found that screening with primary HPV testing generally provided greater reductions in lifetime risk of developing cervical cancer compared with cytology-based screening. A recent study found variation in the utilization of screening tests by health network and geography, with safety-net healthcare systems – which service a greater proportion of uninsured and underinsured women – having a higher use of cytology-based screening than HPV-based screening (*Haas et al., 2022*). Although we did not explicitly simulate screening that switches from cytology-based screening before a disruption to HPV testing upon resumption, such a strategy may be able to mitigate any pandemic-related excess risks. Similarly, we found that the impact of disrupting an HPV-based screening program has different implications than the disruption of a cytology-based program. This can be explained by the fact that HPV screening has a higher sensitivity to detect (pre-invasive) cervical lesions than cytology; therefore, the cancer risk at time of disruption is lower (as there are fewer undetected lesions) and this may provide a greater buffer to endure temporary disruptions. On the other hand, in case of the more sensitive HPV test, disruption takes away a relatively more valuable (i.e., sensitive) screening moment. The balance between these two factors causes a greater or smaller excess risk per delay duration in case of HPV screening compared to cytology screening, which contributes to the within-model differences of cytology-based versus HPV-based screening in *Table 1*. If in a model the first effect (HPV screening contributes to lower risk at the time of disruption) is larger than the second effect (removal of a valuable screening moment), disruption of the HPV program would have a smaller effect than that of the cytology program, which is the case for all screening frequencies in both the Harvard and Policy1-Cervix models and the annual screeners in MISCAN-Cervix (*Table 1*). The MISCAN-Cervix model predicted relatively more excess cancers for women screened with HPV 3-yearly, 5-yearly, or 10-yearly due to disruptions, where delaying a more sensitive test (the second factor) seems to outweigh the first (less underlying disease at the time of the disruption). Differences in dwell time for HPV and cervical precancer among the three models predominately contribute to this balance between the two factors, where the MISCAN-Cervix model has the shortest preclinical dwell time from HPV acquisition to cancer development (*Burger et al., 2020a*). In addition to the shorter dwell times, the MISCAN model also assumes that some precancerous lesions are consistently missed over time by cytology-based screening because they are located deeper into the cervical canal. For women with such lesions, missing a cytology screen due to a disruption is less harmful, which increases the relative difference between primary cytology and primary HPV screening in case of a disruption, and increases the effect of women missing a very sensitive screen (the second factor).

We also found that the relative excess rates of symptomatically detected cancer projected for the same delay period were higher for women overdue for screening compared with women who attend screening according to guidelines. These relative increases were generally similar regardless of the length of the delay period as the underlying cases among the guidelines-compliant and overdue women continued to accumulate with the length of delay. However, some delay-length trends were

observed (*Figure 2*), which may be due to the fact that the differences in the impact of the delay between guidelines-compliant and underscreeners may become smaller in the case of a longer delay (a longer screening delay increasingly becomes more impactful for guidelines-compliant screeners as well). Delay-length differences are predominantly observed in MISCAN-Cervix, which may be in part due to shorter dwell time assumptions. In contrast, for the Harvard and Policy1-Cervix models that assume longer dwell times, the relative impacts between a 1- and 5-year delay are smaller.

Although our analysis was contextualized to the United States, our results may still be generalizable to other countries where cervical screening was disrupted such as the United Kingdom, Ireland, New Zealand, and the Netherlands where cervical screening was paused for 2–4 months (*Public Health Agency, 2020*; *Aitken et al., 2022*; *Campbell et al., 2021*; *National Screening Service, 2022*). We considered combinations of screening modality and screening frequency, some of which will be more applicable to some countries than others. Our overarching findings that disruptions are likely to disproportionately impact those who are already overdue for screening, and that underscreened women are at higher risk than guidelines-compliant screeners affected by a temporary delay are likely generalizable.

Importantly, vaginal HPV-based screening (unlike cytology-based screening) enables self-collection of samples at home, which may provide a tool to reduce screening barriers and facilitate outreach to underscreened people who are also most vulnerable to screening disruptions. In the Netherlands, parts of Sweden, and recently Australia, self-sampling is available to all women, and preliminary findings suggest this has facilitated rapid reintroductions to screening in the Netherlands in the context of COVID-19 (*Aitken et al., 2022*).

## Limitations and clinical relevance

Despite the strength of consistent results from three established CISNET models, there are several limitations that should be considered in interpreting our results. First, in the absence of detailed information on cervical screening disruptions by screening frequency, we explored a range of stylized scenarios that represent different combinations of screening behavior, disruption periods, and screening modalities.

We encountered several challenges as we planned this analysis, which we feel are worth describing as they illustrate some of the limits of modeling in this context. The first is the scope of time over which to consider outcomes. Attempting to assess screening delays over a short period presents problems as short-term outcomes may not be representative of long-term health gains. For instance, the occurrence of screening moments will always be associated with the incidence of cancer due to the volume of screen-detected disease. An analysis that attempted to consider changes in the incidence of cancer within a finite period that includes resumption of screening will generate results that are largely artifacts of the resumption of screening within the period of analysis rather than fundamentally reflecting differences due to temporary extensions to the interval. Therefore, we restricted the short-term analysis to the assessment of symptomatically detected cancers during a finite delay period while the long-term analysis considered the net impact on cervical cancer risk given women's lifetime screening participation.

Considering changes in the longer term required us to make assumptions about women's screening behavior following the COVID-related delays as the impact of the temporary screen delays is contingent on the subsequent screens they receive. Again, we encountered the potential for modeling assumption to influence results. When screening eligibility is limited by an upper age bound, that is, age 65 years, the impact of a screen delay can depend heavily on what is assumed about a woman's final screen (summarized assumptions in *Appendix 1—table 1* and *Appendix 1—table 2*). For instance, a 1-year delay to program for a woman following a 5-year interval would imply a final screen occurring at age 61 rather than age 65 in the 'no delay' scenario. In this instance, the model-projected changes in long-term outcomes potentially reflect a missed final screen moment, rather than isolated only to the extension of 1-year delay (increasing the screening interval temporarily from 5 to 6 years) (*Wang et al., 2017*; *Castanon et al., 2021*). Changes to future screening patterns maybe an overlooked secondary/long-term consequence of this one-time COVID-19 disruption.

This analysis did not consider delays in diagnostic follow-up or precancer treatment, which have been shown to account for most of the potential harm of COVID-induced disruptions to cervical cancer screening (*Burger et al., 2021*; *Smith et al., 2021*). If women who screen less frequently

face further disproportional delays to essential diagnostic services or precancer treatment, we would expect the gap in cancer burden between routine and underscreeners to continue to widen. We did not stratify the impact of temporary disruptions by cancer stage at detection (i.e., potential upstaging due to delayed screening and diagnosis). As COVID-related delays have previously been projected to upstage cancer detection for cervical (*Smith et al., 2021*) and other cancers (*Maringe et al., 2020*), upstaging effects are likely exacerbated for women who screen infrequently, who are already at a higher risk of having a more severe cancer stage at diagnosis (*Landy et al., 2020*). We also did not evaluate rarer outcomes such as cancer death as the impact on incidence is an early indicator for mortality burden. We also did not consider differences in underlying risk between various screening behavior groups (i.e., we assumed differential cancer risk was a function of only screening behavior). If women who screen less frequently also face a higher underlying risk of developing cancer, the differences in our projected risks for delays to underscreeners compared with guideline-compliant screeners may be underestimated, providing additional support for a targeted outreach to under-screened women. Although we did not simulate a cytology and HPV 'co-testing' strategy explicitly, which is a third recommended screening approach by the USPSTF, the majority of the benefit of co-testing has been attributed to the high negative predictive value of HPV testing alone (*Schiffman et al., 2011*); therefore, we expect our current findings for HPV testing alone to be generalizable to women whose last screening involved co-testing.

While our models do not explicitly simulate the impacts of specific factors, including race and ethnicity, poverty income level, education, and insurance status, these characteristics are associated with screening behavior, which we do capture in our simulation models. Furthermore, our projections reflected the burden of cervical cancer assuming an average underlying natural history risk of progression to cervical cancer, and we do not reflect the differential natural history for immunocompromised women. Subsequently, our findings would not be generalizable to certain groups facing greater background risk of developing cancer.

In conclusion, our models predicted that the main driver of lifetime risk of cervical cancer is screening frequency and screening modality rather than temporary disruptions to screening; however, a disruption to screening does not equally impact women with differential screening histories or screening behaviors. Understanding and reaching underscreened women remains the most critical area of focus, regardless of temporary disruptions.

## Materials and methods
### Analytic overview

To complement our previous analysis (*Burger et al., 2021*), we used the same three CISNET-Cervical microsimulation models to project the expected lifetime risk (until age 84 years) of developing cervical cancer for three birth cohorts (born in 1965, 1975, and 1985; aged 55, 45, and 35 in 2020, respectively) assuming an underlying exposure to HPV vaccination (see *Kim et al., 2021*) and screening frequency – that is, annual, 3-yearly, 5-yearly, or 10-yearly screening, aligned so that 2020 was 1, 3, 5, or 10 years since their last screen (*Figure 1*, *Appendix 1—table 1* and *Appendix 1—table 2*). These selected birth cohorts enabled the analysis to capture at least 10 years of pre- and post-COVID-19 screening history. We used the models to estimate both the short- and long-term impacts of COVID-19 delays on cervical cancer burden. As both primary cytology- and HPV-based screening modalities are recommended by the U.S. Preventive Services Task Force (USPSTF), we explored these outcomes in the context of primary cytology (i.e., Pap smear only) and primary HPV-based screening recommendations. USPSTF guidelines recommend cervical cancer screening for women aged 21–65 years with cytology alone every 3 years, with an option to switch to 5-yearly primary high-risk HPV testing from age 30 years with partial genotyping for HPV genotypes 16 and 18 (*Curry et al., 2018*). Screen-positive women (i.e., >= atypical cells of undetermined significance [ASCUS] and reflex high-risk HPV-positive for women following cytology-based screening, or high-risk HPV-positive for women following HPV-based screening) were managed according to guidelines (*Curry et al., 2018*) and followed Kaiser Permanente Northern California compliance patterns, that is, colposcopy compliance (79%), precancer treatment compliance (73%) (*Rendle et al., 2018*). Scenarios were simulated in the context of birth cohort-specific historical HPV vaccination coverage as estimated and applied in other analyses (*Burger et al., 2020b*; *Kim et al., 2021*).

For the short-term impacts, we estimated the relative excess rate of symptomatically detected cancer during a COVID-19-related screening delay period for underscreeners compared with 'guidelines-compliant screeners' (i.e., those who perfectly adhere to 3-year cytology screening or 5-yearly HPV screening, except for during the COVID-19 disruption period). All three models define an incident, symptomatically detected cancer as a cancer that is diagnosed in a woman with a previously undetected cancer who is detected outside routine screening or screening-induced follow-up (e.g., reported symptoms). We averaged the cancer incidence rates (per 100,000 women) accumulated during a delay period (i.e., 1, 2, or 5 years) across the three birth cohorts for each screening history profile (i.e., 3-yearly, 5-yearly, or 10-yearly screening). The denominator for each relative rate (RR) calculation was the accumulated cancer rate under a given delay period for a 'guidelines-compliant' screener, which differed according to primary test modality (i.e., 3-yearly screening for cytology-based screeners and 5-yearly screening for HPV-based screeners) (*Appendix 1—figure 1*). For the long-term impact, we projected the impact of disruptions to lifetime risks and absolute changes in cancer risks for each of the three alternative COVID-19-related screening delay scenarios, compared to a scenario of no COVID-19-related disruptions. To set findings within the wider context of prevention, we additionally considered how much each scenario would reduce a woman's lifetime risk of developing cervical cancer (compared to a hypothetical no screening scenario). For each scenario, model projections of cervical cancer cases and lifetime risks were averaged across the three birth cohorts.

## Simulation models

As previously described (*Burger et al., 2021*; *Burger et al., 2020a*; *de Kok et al., 2020*), the three CISNET-Cervical models (Harvard, MISCAN-Cervix, and Policy1-Cervix) reflect the natural history of HPV-induced cervical cancer but differ structurally with respect to the type and number of health states, HPV genotype categorizations, histological cancer types, model cycle length, and data sources used to parameterize the model prior to fitting to the US population. Standardized US-model inputs included hysterectomy rates, all-cause mortality, and cervical cancer survival (*Burger et al., 2020a*). To reflect the burden of HPV and cervical cancer in the United States, the models were calibrated to HPV and cervical disease outcomes, achieving good fit to empirical targets based on US women (see *Burger et al., 2020a* for details of the calibration and fitting processes).

## Scenarios and assumptions

We assumed that in the absence of the COVID-19 pandemic, each cohort would have received a primary cervical screen in 2020, aligned with an underlying screening frequency, that is, 1, 3, 5, or 10 years since last screen. For each birth cohort and screening frequency combination, these women faced either no delay, or a 1-, 2-, or 5-year delay (*Figure 1*, *Appendix 1—table 2*). We assumed that during the delay period, there was a 100% temporary loss in primary screening, but following the delay period, screening was assumed to immediately resume, and women would continue to follow their pre-pandemic screening frequency. We assumed COVID-19 did not impact attendance for surveillance, diagnosis, or treatment of screen-detected abnormalities or investigation for symptomatically detected cancers, except when directly implied by missed screening events during a COVID-19 delay period. In line with US guideline recommendations, all models assumed women did not attend routine screening after age 65 years. A key modifier of the impact of screening delays on lifetime risk is the age at which women received their last screening test (*Wang et al., 2017*). Due to the analytically fixed screening intervals assumed post-COVID-19-related delay, the timing of future screening was shifted in all cohorts other than annual screeners; as a result, for some combinations of screening frequency and COVID-19-related delays, the delays also reduced the number of lifetime screens and/or changed the age at last screen (see *Appendix 1—table 1* and *Appendix 1—table 2* for additional details). For example, for a woman born in 1975 who screens every 10 years, her last screen would be at age 65 years without a COVID-19 disruption; however, her last screen would occur at age 56, 57, or 60 under the 1-, 2-, or 5-year delay scenarios, respectively.

## Acknowledgements

Emily A Burger receives salary support from the Norwegian Cancer Society (#198073), Megan A Smith receives salary support from the National Health and Medical Research Council, Australia (APP1159491) and Cancer Institute NSW (ECF181561). James O'Mahony is funded by Ireland's Health

Research Board (EIA2017054). Matejka Rebolj receives support from Public Health England, which provided funding for evaluation of various PHE projects; member of various PHE advisory groups for cervical screening; attended meetings with various HPV assay manufacturers; fee for lecture in the last four years from Hologic, paid to employer.

## Additional information

### Competing interests

Karen Canfell: The other authors declare that no competing interests exist.

### Funding

| Funder | Grant reference number | Author |
| --- | --- | --- |
| National Cancer Institute | U01CA199334 | Jane J Kim<br>Karen Canfell<br>Inge MCM de Kok |
| Cancer Research UK | C8162/A27047 | Matejka Rebolj<br>Alejandra Castanon |
| Norwegian Cancer Society | #198073 | Emily A Burger |
| National Health and Medical Research Council, Australia | APP1159491 | Megan A Smith |
| Cancer Institute NSW | ECF181561 | Megan A Smith |
| Ireland's Health Research Board | EIA2017054 | James F O'Mahony |
| Public Health England | | Matejka Rebolj |

The funders had no role in study design, data collection and interpretation, or the decision to submit the work for publication. The contents are solely the responsibility of the authors and do not necessarily represent the official views of the National Cancer Institute or any of the above-mentioned funders.

### Author contributions

Emily A Burger, Conceptualization, Formal analysis, Investigation, Methodology, Visualization, Writing – review and editing; Inge MCM de Kok, Conceptualization, Funding acquisition, Investigation, Methodology, Validation, Writing – review and editing; James F O'Mahony, Conceptualization, Validation, Formal analysis, *Contributed equally as second author; Matejka Rebolj, Sharon J Hanley, Investigation, Methodology, Writing – review and editing; Erik EL Jansen, Software, Investigation, Visualization, Methodology, Formal analysis; Daniel D de Bondt, Software, Investigation, Methodology, Formal analysis; James Killen, Software, Methodology, Formal analysis; Alejandra Castanon, Conceptualization, Validation, Formal analysis; Mary Caroline Regan, Software, Validation, Formal analysis; Jane J Kim, Funding acquisition, Investigation, Methodology, Resources, Writing – review and editing; Karen Canfell, Funding acquisition, Methodology, Project administration, Resources, *Contributed equally as second author; Megan A Smith, Conceptualization, Project administration, Visualization, Methodology, Validation, Formal analysis

### Author ORCIDs

Emily A Burger (ID) http://orcid.org/0000-0002-6657-5849
Matejka Rebolj (ID) http://orcid.org/0000-0001-9597-645X
Erik EL Jansen (ID) http://orcid.org/0000-0003-0436-6918
Sharon J Hanley (ID) http://orcid.org/0000-0002-7554-004X
Megan A Smith (ID) http://orcid.org/0000-0002-0401-2653

### Decision letter and Author response

Decision letter https://doi.org/10.7554/eLife.81711.sa1
Author response https://doi.org/10.7554/eLife.81711.sa2

# Additional files

## Supplementary files
• MDAR checklist

## Data availability

Supporting Information contained in the Supplementary Material of Burger et al. 2020a provides details on microsimulation model inputs, calibration to epidemiologic data, and calibration approach in line with good modeling practice. This study involved modelling rather than direct analysis of primary datasets. The current manuscript is a computational study, so no data have been generated for this manuscript. The Cancer Intervention and Surveillance Modeling Network (CISNET) (https://cisnet.cancer.gov/) Cervix model codes have been developed over decades, are proprietary property, and cannot be provided by the authors at this time; however, CISNET-Cervix, under our CISNET 'Model Accessibility' interest group is working to provide transparent and reproducible modeling code for forthcoming projects ("C4"). Access to current code is possible only through supervised training at each modeling group site.

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

# Appendix 1

## Analytic screening assumptions

There were differences in how each modeling group applied screening. In order to isolate health outcomes for women disrupted in 2020, the Harvard and MISCAN-Cervix models assumed primary screening followed fixed intervals (so-called 'age-based' screening) irrespective of possible follow-up testing, which would normally affect the age at which a woman would next screen. In contrast, the Policy1-Cervix model allowed for dynamic primary screening based on time since last primary or follow-up screen, but isolated their model outcomes to the women who were delayed in 2020, that is, did not have previous positive results. Subsequently, the Policy1-Cervix model reflects a marginally lower-risk cohort of women; however, while the absolute lifetime risk may be lower compared with the other two models, absolute changes in risk compared with their counterfactual of 'no delay' will isolate the impact of delays.

In line with US guideline recommendations, all models assumed screening ended no later than age 65 years (inclusive). Scenarios involving delays to screening shifted the timing of future screening in all groups other than annual screeners and/or some screening frequency and delay combinations, the delays to screening reduced the number of lifetime screens and changed the implied age of last screen (due to fixed screening intervals post-delay) (*Appendix 1—table 1* and *Appendix 1—table 2*). For example, for 10-yearly screeners in the 1975 birth cohort (age 45 in 2020), a 1-year delay meant their last screening test occurred at age 56, 9 years earlier than in the 'no delay' scenario. In these same 10-yearly screeners, a 2- and 5-year delay shifted their age at last screen to age 57 and 60 years, respectively. Some of the added risks of the longer delay compared with a 1-year delay may be mitigated by the later screening end age in the longer delay scenarios (that end screening at age 57 or 60, compared to age 56). Model-based analyses require making analytic assumptions about imperfect screening behavior and guidelines.

**Appendix 1—table 1.** Screening end age (lifetime number of screens) by birth cohort, screening frequency, and delay duration.

| | | Annual | 3-yearly* | 5-yearly† | 10-yearly |
|---|---|---|---|---|---|
| | No delay | 65 (45) | 64 (15) | 65 (9) | 65 (5) |
| | 1-year delay | 65 (44) | 65 (15) | 61 (8) | 56 (4) |
| | 2-year delay | 65 (43) | 63 (14) | 62 (8) | 57 (4) |
| 1965 (age 55 in 2020) | 5-year delay | 65 (40) | 63 (13) | 65 (8) | 60 (4) |
| | No delay | 65 (45) | 63 (15) | 65 (9) | 65 (5) |
| | 1-year delay | 65 (44) | 64 (15) | 61 (8) | 56 (4) |
| | 2-year delay | 65 (43) | 65 (15) | 62 (8) | 57 (4) |
| 1975 (age 45 in 2020) | 5-year delay | 65 (40) | 65 (14) | 65 (8) | 60 (4) |
| | No delay | 65 (45) | 65 (15) | 65 (8) | 65 (5) |
| | 1-year delay | 65 (44) | 63 (14) | 61 (8) | 56 (4) |
| | 2-year delay | 65 (43) | 64 (14) | 62 (8) | 57 (4) |
| 1985 (age 35 in 2020) | 5-year delay | 65 (40) | 64 (13) | 65 (8) | 60 (4) |

*Guidelines-compliant screener with primary cytology-based screening.
†Guidelines-compliant screener with primary human papillomavirus (HPV) testing for women aged 30+ years.

**Appendix 1—table 2.** Example age at screen for the 1975 birth cohort without (highlighted in green) and with (highlighted in yellow) COVID-19-related delays by screening frequency.
Numbers under each delay are ages, bolded numbers are ages at which screening takes place, green highlight reflects no delay, and yellow highlights reflects a delay.

| No delay | | | | | 1-year delay | | | | | 2-year delay | | | | | 5-year delay | | | | |
|---|---|---|---|---|---|---|---|---|---|---|---|---|---|---|---|---|---|---|---|---|
| Year | Q1 | Q3 | Q5 | Q10 | Year | Q1 | Q3 | Q5 | Q10 | Year | Q1 | Q3 | Q5 | Q10 | Year | Q1 | Q3 | Q5 | Q10 |
| 1996 | **21** | **21** | 21 | 21 | 1996 | **21** | **21** | 21 | 21 | 1996 | **21** | **21** | 21 | 21 | 1996 | **21** | **21** | 21 | 21 |
| 1997 | **22** | 22 | 22 | 22 | 1997 | **22** | 22 | 22 | 22 | 1997 | **22** | 22 | 22 | 22 | 1997 | **22** | 22 | 22 | 22 |

*Appendix 1—table 2 Continued on next page*

*Appendix 1—table 2 Continued*

| | No delay | | | | | 1-year delay | | | | | 2-year delay | | | | | 5-year delay | | | |
|---|---|---|---|---|---|---|---|---|---|---|---|---|---|---|---|---|---|---|---|
| 1998 | 23 | 23 | 23 | 23 | 1998 | 23 | 23 | 23 | 23 | 1998 | 23 | 23 | 23 | 23 | 1998 | 23 | 23 | 23 | 23 |
| 1999 | 24 | 24 | 24 | 24 | 1999 | 24 | 24 | 24 | 24 | 1999 | 24 | 24 | 24 | 24 | 1999 | 24 | 24 | 24 | 24 |
| 2000 | 25 | 25 | 25 | 25 | 2000 | 25 | 25 | 25 | 25 | 2000 | 25 | 25 | 25 | 25 | 2000 | 25 | 25 | 25 | 25 |
| 2001 | 26 | 26 | 26 | 26 | 2001 | 26 | 26 | 26 | 26 | 2001 | 26 | 26 | 26 | 26 | 2001 | 26 | 26 | 26 | 26 |
| 2002 | 27 | 27 | 27 | 27 | 2002 | 27 | 27 | 27 | 27 | 2002 | 27 | 27 | 27 | 27 | 2002 | 27 | 27 | 27 | 27 |
| 2003 | 28 | 28 | 28 | 28 | 2003 | 28 | 28 | 28 | 28 | 2003 | 28 | 28 | 28 | 28 | 2003 | 28 | 28 | 28 | 28 |
| 2004 | 29 | 29 | 29 | 29 | 2004 | 29 | 29 | 29 | 29 | 2004 | 29 | 29 | 29 | 29 | 2004 | 29 | 29 | 29 | 29 |
| 2005 | 30 | 30 | 30 | 30 | 2005 | 30 | 30 | 30 | 30 | 2005 | 30 | 30 | 30 | 30 | 2005 | 30 | 30 | 30 | 30 |
| 2006 | 31 | 31 | 31 | 31 | 2006 | 31 | 31 | 31 | 31 | 2006 | 31 | 31 | 31 | 31 | 2006 | 31 | 31 | 31 | 31 |
| 2007 | 32 | 32 | 32 | 32 | 2007 | 32 | 32 | 32 | 32 | 2007 | 32 | 32 | 32 | 32 | 2007 | 32 | 32 | 32 | 32 |
| 2008 | 33 | 33 | 33 | 33 | 2008 | 33 | 33 | 33 | 33 | 2008 | 33 | 33 | 33 | 33 | 2008 | 33 | 33 | 33 | 33 |
| 2009 | 34 | 34 | 34 | 34 | 2009 | 34 | 34 | 34 | 34 | 2009 | 34 | 34 | 34 | 34 | 2009 | 34 | 34 | 34 | 34 |
| 2010 | 35 | 35 | 35 | 35 | 2010 | 35 | 35 | 35 | 35 | 2010 | 35 | 35 | 35 | 35 | 2010 | 35 | 35 | 35 | 35 |
| 2011 | 36 | 36 | 36 | 36 | 2011 | 36 | 36 | 36 | 36 | 2011 | 36 | 36 | 36 | 36 | 2011 | 36 | 36 | 36 | 36 |
| 2012 | 37 | 37 | 37 | 37 | 2012 | 37 | 37 | 37 | 37 | 2012 | 37 | 37 | 37 | 37 | 2012 | 37 | 37 | 37 | 37 |
| 2013 | 38 | 38 | 38 | 38 | 2013 | 38 | 38 | 38 | 38 | 2013 | 38 | 38 | 38 | 38 | 2013 | 38 | 38 | 38 | 38 |
| 2014 | 39 | 39 | 39 | 39 | 2014 | 39 | 39 | 39 | 39 | 2014 | 39 | 39 | 39 | 39 | 2014 | 39 | 39 | 39 | 39 |
| 2015 | 40 | 40 | 40 | 40 | 2015 | 40 | 40 | 40 | 40 | 2015 | 40 | 40 | 40 | 40 | 2015 | 40 | 40 | 40 | 40 |
| 2016 | 41 | 41 | 41 | 41 | 2016 | 41 | 41 | 41 | 41 | 2016 | 41 | 41 | 41 | 41 | 2016 | 41 | 41 | 41 | 41 |
| 2017 | 42 | 42 | 42 | 42 | 2017 | 42 | 42 | 42 | 42 | 2017 | 42 | 42 | 42 | 42 | 2017 | 42 | 42 | 42 | 42 |
| 2018 | 43 | 43 | 43 | 43 | 2018 | 43 | 43 | 43 | 43 | 2018 | 43 | 43 | 43 | 43 | 2018 | 43 | 43 | 43 | 43 |
| 2019 | 44 | 44 | 44 | 44 | 2019 | 44 | 44 | 44 | 44 | 2019 | 44 | 44 | 44 | 44 | 2019 | 44 | 44 | 44 | 44 |
| 2020 | 45 | 45 | 45 | 45 | 2020 | 45 | 45 | 45 | 45 | 2020 | 45 | 45 | 45 | 45 | 2020 | 45 | 45 | 45 | 45 |
| 2021 | 46 | 46 | 46 | 46 | 2021 | 46 | 46 | 46 | 46 | 2021 | 46 | 46 | 46 | 46 | 2021 | 46 | 46 | 46 | 46 |
| 2022 | 47 | 47 | 47 | 47 | 2022 | 47 | 47 | 47 | 47 | 2022 | 47 | 47 | 47 | 47 | 2022 | 47 | 47 | 47 | 47 |
| 2023 | 48 | 48 | 48 | 48 | 2023 | 48 | 48 | 48 | 48 | 2023 | 48 | 48 | 48 | 48 | 2023 | 48 | 48 | 48 | 48 |
| 2024 | 49 | 49 | 49 | 49 | 2024 | 49 | 49 | 49 | 49 | 2024 | 49 | 49 | 49 | 49 | 2024 | 49 | 49 | 49 | 49 |
| 2025 | 50 | 50 | 50 | 50 | 2025 | 50 | 50 | 50 | 50 | 2025 | 50 | 50 | 50 | 50 | 2025 | 50 | 50 | 50 | 50 |
| 2026 | 51 | 51 | 51 | 51 | 2026 | 51 | 51 | 51 | 51 | 2026 | 51 | 51 | 51 | 51 | 2026 | 51 | 51 | 51 | 51 |
| 2027 | 52 | 52 | 52 | 52 | 2027 | 52 | 52 | 52 | 52 | 2027 | 52 | 52 | 52 | 52 | 2027 | 52 | 52 | 52 | 52 |
| 2028 | 53 | 53 | 53 | 53 | 2028 | 53 | 53 | 53 | 53 | 2028 | 53 | 53 | 53 | 53 | 2028 | 53 | 53 | 53 | 53 |
| 2029 | 54 | 54 | 54 | 54 | 2029 | 54 | 54 | 54 | 54 | 2029 | 54 | 54 | 54 | 54 | 2029 | 54 | 54 | 54 | 54 |
| 2030 | 55 | 55 | 55 | 55 | 2030 | 55 | 55 | 55 | 55 | 2030 | 55 | 55 | 55 | 55 | 2030 | 55 | 55 | 55 | 55 |
| 2031 | 56 | 56 | 56 | 56 | 2031 | 56 | 56 | 56 | 56 | 2031 | 56 | 56 | 56 | 56 | 2031 | 56 | 56 | 56 | 56 |
| 2032 | 57 | 57 | 57 | 57 | 2032 | 57 | 57 | 57 | 57 | 2032 | 57 | 57 | 57 | 57 | 2032 | 57 | 57 | 57 | 57 |
| 2033 | 58 | 58 | 58 | 58 | 2033 | 58 | 58 | 58 | 58 | 2033 | 58 | 58 | 58 | 58 | 2033 | 58 | 58 | 58 | 58 |
| 2034 | 59 | 59 | 59 | 59 | 2034 | 59 | 59 | 59 | 59 | 2034 | 59 | 59 | 59 | 59 | 2034 | 59 | 59 | 59 | 59 |
| 2035 | 60 | 60 | 60 | 60 | 2035 | 60 | 60 | 60 | 60 | 2035 | 60 | 60 | 60 | 60 | 2035 | 60 | 60 | 60 | 60 |
| 2036 | 61 | 61 | 61 | 61 | 2036 | 61 | 61 | 61 | 61 | 2036 | 61 | 61 | 61 | 61 | 2036 | 61 | 61 | 61 | 61 |
| 2037 | 62 | 62 | 62 | 62 | 2037 | 62 | 62 | 62 | 62 | 2037 | 62 | 62 | 62 | 62 | 2037 | 62 | 62 | 62 | 62 |
| 2038 | 63 | 63 | 63 | 63 | 2038 | 63 | 63 | 63 | 63 | 2038 | 63 | 63 | 63 | 63 | 2038 | 63 | 63 | 63 | 63 |
| 2039 | 64 | 64 | 64 | 64 | 2039 | 64 | 64 | 64 | 64 | 2039 | 64 | 64 | 64 | 64 | 2039 | 64 | 64 | 64 | 64 |
| 2040 | 65 | 65 | 65 | 65 | 2040 | 65 | 65 | 65 | 65 | 2040 | 65 | 65 | 65 | 65 | 2040 | 65 | 65 | 65 | 65 |
| 2041 | 66 | 66 | 66 | 66 | 2041 | 66 | 66 | 66 | 66 | 2041 | 66 | 66 | 66 | 66 | 2041 | 66 | 66 | 66 | 66 |

*Modeling outputs provided cancer rates on an annual (yearly) basis; therefore, a small proportion of 'delay-related' cases are not captured (e.g., a one-year disruption that started March 2020 would continue into February 2021). Our model-based outputs would, therefore, underestimate a small proportion of cancer-burden; however, these underestimates would be captured similarly in both the numerator and denominator of the RR, likely underestimating the RR). Abbreviations: Qx, primary screening interval; dx, delay period; X, a new screen occurs

**Appendix 1—figure 1.** Schematic of short-term cancer burden calculations*.

## Additional results

**Appendix 1—table 3.** Relative rate ratios and accumulated incidence rates per 100,000 women for each screening frequency and delay scenario.

| Screening frequency | 1-year delay | 2-year delay | 5-year delay |
|---|---|---|---|
| *Policy1-Cervix* | | | |
| 5-yearly screener (cytology) | 1.40 = 4.94/3.58 | 1.44 = 10.05/7.00 | 1.57 = 30.03/19.10 |
| 10-yearly screener (cytology) | 3.10 = 11.06/3.58 | 3.35 = 23.45/7.00 | 3.34 = 63.72/19.10 |
| 10-yearly screener (HPV) | 2.26 = 7.36/3.26 | 2.48 = 15.31/6.18 | 2.54 = 42.41/16.67 |
| *MISCAN-Cervix* | | | |
| 5-yearly screener (cytology) | 2.78 = 3.00/1.08 | 2.28 = 6.41/2.81 | 1.99 = 22.09/11.08 |
| 10-yearly screener (cytology) | 6.97 = 7.54/1.08 | 5.73 = 16.11/2.81 | 4.15 = 45.97/11.08 |
| 10-yearly screener (HPV) | 3.91 = 6.08/1.55 | 3.67 = 13.08/3.56 | 2.97 = 38.84/13.09 |
| *Harvard* | | | |
| 5-yearly screener (cytology) | 1.8 = 9.64/5.37 | 1.74 = 17.69/10.15 | 1.64 = 45.97/27.96 |
| 10-yearly screener (cytology) | 3.8 = 20.39/5.37 | 3.64 = 36.94/10.15 | 3.19 = 89.29/27.96 |
| 10-yearly screener (HPV) | 3.61 = 10.74/2.98 | 3.58 = 19.51/5.45 | 3.69 = 46.83/12.68 |

Relative rate ratio is calculated as the accumulated incidence rate per 100,000 women during a delay period for a give screening history divided by the accumulated incidence rate per 100,000 women during the same delay period among guidelines-compliant screeners. Incidence rates are the average across the three birth cohorts. 3-yearly cytology screening is considered guidelines-compliant screening; 5-yearly human papillomavirus (HPV) screening is considered guidelines compliant.

**Appendix 1—table 4.** Percentage reduction in average (across the 1965, 1975, and 1985 birth cohorts) lifetime risk of cancer compared with no screening.

| | Screening frequency | | | |
|---|---|---|---|---|
| | Annual (%) | 3-yearly (%) | 5-yearly (%) | 10-yearly (%) |
| Primary cytology-based screening | | | | |
| *Harvard* | | | | |
| No delay | 88.4 | 79.9 | 71.4 | 56.3 |
| 1-year delay | 88.5 | 80.0 | 70.9 | 55.3 |
| 2-year delay | 88.4 | 79.5 | 70.5 | 54.9 |

*Appendix 1—table 4 Continued on next page*

*Appendix 1—table 4 Continued*

| | Screening frequency | | | |
|---|---|---|---|---|
| | **Annual (%)** | **3-yearly (%)** | **5-yearly (%)** | **10-yearly (%)** |
| 5-year delay | 87.9 | 78.4 | 69.3 | 53.4 |
| *MISCAN-Cervix* | | | | |
| No delay | 85.1 | 72.1 | 62.1 | 48.4 |
| 1-year delay | 85.0 | 71.7 | 60.2 | 44.7 |
| 2-year delay | 84.7 | 71.2 | 59.9 | 44.9 |
| 5year delay | 83.1 | 69.1 | 58.8 | 43.9 |
| *Policy1-Cervix* | | | | |
| No delay | 87.7 | 86.5 | 84.5 | 75.9 |
| 1-year delay | 87.8 | 86.3 | 82.3 | 71.9 |
| 2-year delay | 87.7 | 85.8 | 82.5 | 71.9 |
| 5-year delay | 86.8 | 85.1 | 82.5 | 71.3 |
| **Primary HPV-based screening** | | | | |
| *Harvard* | | | | |
| No delay | 92.6 | 89.3 | 86.7 | 77.8 |
| 1-year delay | 92.6 | 89.4 | 86.6 | 77.1 |
| 2-year delay | 92.6 | 89.4 | 86.6 | 76.7 |
| 5-year delay | 92.6 | 89.2 | 86.4 | 75.0 |
| *MISCAN-Cervix* | | | | |
| No delay | 94.0 | 84.2 | 74.8 | 58.0 |
| 1-year delay | 93.9 | 83.8 | 72.3 | 53.5 |
| 2-year delay | 93.6 | 83.2 | 72.3 | 53.5 |
| 5-year delay | 92.3 | 81.0 | 70.8 | 51.8 |
| *Policy1-Cervix* | | | | |
| No delay | 90.9 | 90.4 | 90.1 | 86.6 |
| 1-year delay | 90.9 | 90.4 | 88.7 | 82.7 |
| 2-year delay | 91.0 | 90.1 | 89.0 | 82.8 |
| 5-year delay | 91.0 | 90.0 | 89.2 | 82.6 |

HPV, human papillomavirus.

