## [Editor Report]

This article describes the use of three well‐established mathematical models of cervical cancer to estimate the impact of COVID‐19-related delays in screening access on cervical cancer incidence and delays in diagnosis. Consistent with previous work and the known biology of cervical cancers, the findings that short delays have relatively small effects on population‐level cervical cancer risk are reassuring overall, but the impact of screening interval and screening test performance suggests that existing disparities related to screening access may be exacerbated. These results should be useful for policymakers in planning responses to future pandemics or other sources of sudden restriction of screening availability.

---

## [Decision Letter]

**Decision letter after peer review:**

Thank you for submitting your article "Health impacts of COVID-19 disruptions to primary cervical screening by time since last screen: A model-based analysis for current and future disruptions" for consideration by *eLife*. Your article has been reviewed by 2 peer reviewers, and the evaluation has been overseen by a Reviewing Editor and a Senior Editor. The following individual involved in review of your submission has agreed to reveal their identity: Evan Myers (Reviewer #1).

As is customary in *eLife*, the reviewers have discussed their critiques with one another. What follows below is the Reviewing Editor's compilation of the essential and ancillary points provided by reviewers in their critiques and in their interaction post-review. Please submit a revised version that addresses these concerns directly. Although we expect that you will address these comments in your response letter, we also need to see the corresponding revision clearly marked in the text of the manuscript. Some of the reviewers' comments may seem to be simple queries or challenges that do not prompt revisions to the text. Please keep in mind, however, that readers may have the same perspective as the reviewers. Therefore, it is essential that you attempt to amend or expand the text to clarify the narrative accordingly.

Essential revisions:

1. The model assumes there are no delays in diagnostic follow-up services and that the time to diagnosis is equally distributed across categories of screeners. Some further discussion on this assumption and its impact on predictions may be warranted.

2. Further clarification on how symptomatic cancers are defined and modeled in each model should be provided.

3. There should be a brief description of the US cervical cancer screening guidelines that are being modeled for readers who may be unfamiliar with these guidelines.

*Reviewer #1 (Recommendations for the authors):*

I don't have any major substantive comments/suggestions. As noted in my public review, it might be worth a little discussion about stage shift, and the potential for factors affecting underscreening also leading to delays in diagnosis after development of screening.

*Reviewer #2 (Recommendations for the authors):*

The authors have written an extremely informative and comprehensive modeling manuscript showing the import of various screening factors on the incidence of cervical cancer in a population, in the short term and over a lifetime. For the most part the paper was easy to follow but may benefit from a few clarifications and points of consideration.

This paper was based on the US population. Co-testing every 5 years is also a standard there, and Kaiser Permanente (referenced for data on abnormal test follow up compliance) has published its long standing co-test data. It may be worthwhile in the background to comment that this strategy was not modeled and found to be non cost effective (or whatever reason for not modeling).

The outcome was symptomatic cervical cancer. A comment or explanation as to what symptomatic cervical cancer is for model detection would be helpful. The individual models have detailed appendices as does the CISNET summary, but within the text would help the reader understand the difference between screen detected, interval and symptomatic cancer. Later in the discussion screening moments and screen detected cancer are discussed.

As noted in the public review, greater explanation of the US cervical screening guidelines would help the reader understand the transition at 30 years to HPV from cytology. The authors described that disadvantaged women could not screen regularly, whereas women with insurance could screen more frequently ie annually. For a non US reader, is cytology screening, being an inferior test to HPV, as shown in the modeling results, available to un-insured or poorer disadvantaged women?

In the discussion, please label or clearly identify the two factors discussed for "first" and "second effect". Are they better HPV test sensitivity and NPV for factor 1, and screening moment or opportunity for factor 2? "The balance between these two factors causes a greater or smaller excess risk per delay duration in case of HPV screening compared to cytology screening." The sentence is somewhat confusing. Maybe saying HPV screening is superior to cytology but the benefit varies depending on the balance of factors? The rest of the paragraph explains some of the varying results among the 3 models and the explanation is done well referring back to the shortened dwell time in the Miscan model.

Overall, I very much enjoyed reading this manuscript.

---

## [Author Response]

Essential revisions:1. The model assumes there are no delays in diagnostic follow-up services and that the time to diagnosis is equally distributed across categories of screeners. Some further discussion on this assumption and its impact on predictions may be warranted.

We agree, and the reviewers raise an important distinction between this analysis and our previous analyses, which showed that most of the disruption-induced harm is due to delays in diagnosis and treatment services (Burger et al., *Journal of Medical Screening* 2021/Smith et al., *Preventive Medicine* 2021), provided that women are routinely screened. Any further delays for women who screen infrequently would likely continue to widen the gap in cervical cancer burden between women who access screening infrequently and routinely.

On page 8 of the non-tracked changes version, we have added,

“This analysis did not consider delays in diagnostic follow-up or precancer treatment, which have been shown to account for most of the potential harm of COVID-induced disruptions to cervical cancer screening (14, 15). If women who screen less frequently face further and disproportional delays to essential diagnostic services or precancer treatment, we would expect the gap in cancer burden between routine and under-screeners to continue to widen.”

2. Further clarification on how symptomatic cancers are defined and modeled in each model should be provided.

All three models define symptomatic cancers similarly, which we have now defined in the Methods section.

On page 9 we have added,

“All three models define an incident, symptomatically-detected cancer as a cancer that is diagnosed in a woman with a previously undetected cancer who is detected outside routine screening or screening-induced follow-up (e.g., reported symptoms).”

3. There should be a brief description of the US cervical cancer screening guidelines that are being modeled for readers who may be unfamiliar with these guidelines.

We have added more detail to the guidelines descriptions on page 9.

“As both primary cytology- and HPV-based screening modalities are recommended by the U.S. Preventive Services Task Force (USPSTF), we explored these outcomes in the context of primary cytology (i.e., Pap smear only) and primary HPV-based screening recommendations. USPSTF guidelines recommend cervical cancer screening for women aged 21-65 years with cytology alone every three years, with an option to switch to 5-yearly primary high-risk HPV testing from age 30 years with partial genotyping for HPV genotypes 16 and 18 (17). Screen-positive women (i.e., atypical cells of undetermined significance (ASCUS) and reflex high-risk HPV-positive for women following cytology-based screening, or high-risk HPV-positive for women following HPV-based screening) were managed according to guidelines (17) and followed Kaiser Permanente Northern California compliance….”

Reviewer #1 (Recommendations for the authors):I don't have any major substantive comments/suggestions. As noted in my public review, it might be worth a little discussion about stage shift, and the potential for factors affecting underscreening also leading to delays in diagnosis after development of screening.

Thank you for the positive feedback. In addition to the changes addressed above in the Essential Revisions, we have also added to our discussion on page 8,

“We did not stratify the impact of temporary disruptions by cancer stage at detection (i.e., potential upstaging due to delayed screening and diagnosis). As COVID-related delays have previously been projected to upstage cancer detection for cervical (Smith et al., 2021) and other cancers (Maringe et al., 2020), upstaging effects are likely exacerbated for women who screen infrequently, who are already at a higher risk of having a more severe cancer stage at diagnosis (5). We also did not evaluate rarer outcomes such as cancer death….”

Reviewer #2 (Recommendations for the authors):The authors have written an extremely informative and comprehensive modeling manuscript showing the import of various screening factors on the incidence of cervical cancer in a population, in the short term and over a lifetime. For the most part the paper was easy to follow but may benefit from a few clarifications and points of consideration.This paper was based on the US population. Co-testing every 5 years is also a standard there, and Kaiser Permanente (referenced for data on abnormal test follow up compliance) has published its long standing co-test data. It may be worthwhile in the background to comment that this strategy was not modeled and found to be non cost effective (or whatever reason for not modeling).

Thank you for reminder to justify our exclusion of a co-testing strategy. On page 8, we added,

“Although we did not simulate a cytology and HPV “co-testing” strategy explicitly, which is a third recommended screening approach by the USPSTF, the majority of the benefit of co-testing has been attributed to the high negative predictive value of HPV testing alone (Schiffman et al., 2011); therefore, we expect our current findings for HPV testing alone to be generalizable to women whose last screening involved co-testing.”

The outcome was symptomatic cervical cancer. A comment or explanation as to what symptomatic cervical cancer is for model detection would be helpful. The individual models have detailed appendices as does the CISNET summary, but within the text would help the reader understand the difference between screen detected, interval and symptomatic cancer. Later in the discussion screening moments and screen detected cancer are discussed.

We have addressed this above in the Essential Revision section, and on page 5 of the manuscript.

As noted in the public review, greater explanation of the US cervical screening guidelines would help the reader understand the transition at 30 years to HPV from cytology. The authors described that disadvantaged women could not screen regularly, whereas women with insurance could screen more frequently ie annually. For a non US reader, is cytology screening, being an inferior test to HPV, as shown in the modeling results, available to un-insured or poorer disadvantaged women?

Given the USPSTF recommendations that include both cytology and primary HPV testing, both screening modalities are now readily available throughout the US. However, variation exists in the utilization of screening tests by health network and geography, with safety-net health care systems – which service a greater proportion of uninsured and underinsured women – having a higher use of Pap testing than HPV testing (Haas et al.). The National Breast and Cervical Cancer Early Detection Program, a governmental program run by the US CDC that provides breast and cervical cancer screening to low-income women, also documents a high proportion Pap testing usage (https://www.cdc.gov/cancer/nbccedp/data/summaries/).

To the Discussion on page 6 we added,

“A recent study found variation in the utilization of screening tests by health network and geography, with safety-net health care systems – which service a greater proportion of uninsured and underinsured women – having a higher use of Pap testing than HPV testing (Haas et al., 2022).”

In the discussion, please label or clearly identify the two factors discussed for "first" and "second effect". Are they better HPV test sensitivity and NPV for factor 1, and screening moment or opportunity for factor 2? "The balance between these two factors causes a greater or smaller excess risk per delay duration in case of HPV screening compared to cytology screening." The sentence is somewhat confusing. Maybe saying HPV screening is superior to cytology but the benefit varies depending on the balance of factors? The rest of the paragraph explains some of the varying results among the 3 models and the explanation is done well referring back to the shortened dwell time in the Miscan model.

We have addressed your comment above.

Overall, I very much enjoyed reading this manuscript.

Thank you, again.